# Disability during Early Pregnancy: Using the Sheehan Disability Scale during the First Trimester in Japan

**DOI:** 10.3390/healthcare10122514

**Published:** 2022-12-12

**Authors:** Ayako Hada, Mariko Minatani, Mikiyo Wakamatsu, Toshinori Kitamura

**Affiliations:** 1Kitamura Institute of Mental Health Tokyo, Tokyo 151-0063, Japan; 2Kitamura KOKORO Clinic Mental Health, Tokyo 151-0063, Japan; 3Department of Community Mental Health & Law, National Institute of Mental Health, National Center of Neurology and Psychiatry, Tokyo 187-8553, Japan; 4Life Value Creation Unit, NTT DATA Institute of Management Consulting, Inc., Tokyo 102-0093, Japan; 5Department of Reproductive Health Care Nursing, Kagoshima University Faculty of Medicine School of Health Sciences, Kagoshima 890-8544, Japan; 6T. and F. Kitamura Foundation for Studies and Skill Advancement in Mental Health, Tokyo 151-0063, Japan; 7Department of Psychiatry, Graduate School of Medicine, Nagoya University, Nagoya 466-8550, Japan

**Keywords:** Sheehan Disability Scale, factor structure, measurement and structural invariance, parity, disability in pregnancy

## Abstract

**Background:** Many pregnant women experience impairments in social, occupational, or other important functioning. **Aim:** This study aimed to confirm measurement and structural invariance of the Sheehan Disability Scale (SDS) and its validity during early pregnancy. **Design:** Longitudinal study with two observations. **Methods:** Questionnaires were distributed to pregnant women attending antenatal clinics at gestational weeks 10–13. Of 382 respondents, 129 responded to the SDS again 1 week later. **Results:** Confirmatory factor analysis shows good fit with the data: *χ*^2^/*df* = 0, comparative fit index (CFI) = 1.000, standardized root mean square residual (SRMR) = 0, and root mean square error of approximation (RMSEA) = 0.718. There is acceptable configural, measurement, and structural invariance of the factor structure between primiparas and multiparas as well as between two observation occasions. The Pregnancy–Unique Quantification of Emesis and Nausea, Patient Health Questionnaire-9, and Insomnia Severity Index subscales explain 47% of the variance in SDS scores. **Conclusion:** Perinatal health care professionals should pay more attention to the difficulties and disabilities that pregnant women face.

## 1. Introduction

Pregnancy used to be considered to involve little or no psychological or physical difficulties. Disabilities during pregnancy have not received much attention in clinical and research settings. For example, a study in Hamilton County over a period of 18 years shows a psychiatric admission rate of 40.3 per 10,000 women-years among women within 6 months after childbirth, which is much higher than that among non-childbearing women (35.1 per 10,000 women-years) [1]. Surprisingly, this rate is lower than the rate among pregnant women (7.1 per 10,000 women-years). The argument that pregnancy is a psychologically stable period based only on the admission rate may be spurious and paradoxical because psychiatric admission is only an indicator of severe mental illness. Pregnant women, even when psychologically ill, might be more likely to receive care at home. A recent epidemiological study shows that, during the ante-natal versus post-natal periods, women are at similar risk of developing mental illness, mainly mood and anxiety disorders [2]. Approximately 12% of women had an onset of a psychiatric disorder during pregnancy. Another 12% had an onset of a psychiatric disorder during the 3 month period after childbirth. Mental illnesses during pregnancy include depression, obsessive–compulsive disorder [3], sleep disorders [4], other anxiety disorders, eating disorders [5], and tokophobia [6,7], among others [8]. Approximately 28% to 38% of pregnant women experience sleep deficiency during early pregnancy, which includes delayed sleep onset (difficulty falling asleep), frequent awakening during the night, early morning awakening, and poor sleep quality [9]. These sleep problems often are followed by daytime symptoms such as tiredness, poor concentration, and irritability.

Pregnancy is also characterized by difficulties related to physical problems such as nausea and vomiting. Almost 70% of women worldwide experience nausea and vomiting of pregnancy (NVP) [10]. Among women with NVP, the emergence of physical symptoms impairs their ability to function [11,12]. NVP has significant effects on the ability to continue performing daily activities and work capacity [13,14]. During the normal progression of pregnancy, pregnant women experience pregnancy-related symptoms such as tiredness, heartburn, backache, and headache [15,16]. These physical discomforts may cause restrictions in daily activity.

The functional difficulties described above frequently limit the ability of pregnant women to engage in important tasks and participate in daily activities. Many pregnant women experience impairments in social (e.g., leisure activity and relating with friends), occupational (e.g., house chores and looking after children), or other important areas of functioning. Although reduced functioning might not be caused by major physical or psychiatric illness, pregnant women might experience poor quality of life due to common mental or physical ill conditions such as antenatal depression [2], NVP [14,17,18], and insomnia [18]. Whatever the cause, impaired functioning should be seen as a disability. Such disabilities are prevalent in work, study, family life, home responsibilities, social life, and leisure activities. Those impaired functions or functional difficulties were dealt with as burdens [14], and were also recognized as correlates of adverse pregnancy outcomes [9].

Effects of such disabilities on pregnancy and neonatal outcomes have not been extensively studied, despite the fact that mental and physical ill-health affect these conditions (e.g., [18,19,20,21,22]). To further our knowledge and to advance nursing care, we need a reliable, valid, and simple-to-use measure of disability in pregnant women. The Sheehan Disability Scale (SDS; @Copyright 1983–2020 Sheehan DV. All right reserved. May be reproduced only with the permission of Dr. David, V., Sheehan, copyright holder. For permission contact davidsheehan@gmail.com) [23] is a disability measure used in epidemiological and treatment outcome studies [24,25,26,27,28,29]. The SDS contains three items, which is much shorter than very complex assessments of disability such as the International Classification of Impairments, Disabilities, and Handicaps (ICIDH) [30]. The Social Adjustment Scale–Self Report [31] is widely used, but it has 42 items. The SDS covers (a) work and schoolwork, (b) social and leisure activities, and (c) family life and home responsibilities. The SDS is brief and simple to rate. It is also sensitive to changes in the patient’s clinical status [32]. It has test–retest reliability and concurrent validity [33]. Despite its potential clinical utility, the SDS has rarely been administered to women during the perinatal period (e.g., [34]), and it has never been used in pregnant women.

This study evaluates the psychometric properties of the SDS among pregnant women, including goodness-of-fit and configural, measurement, and structural invariance of the factor structure. The study focuses on the invariance (stability) of the factor structure between nulliparas and multiparas and between two observations because the selection of the best-fit model of the factor structure cannot assume that the psychological instrument in question measures the same phenomena when used in different populations, or used in the same population but on more than one measurement occasion. If not, indicators of the instrument do not have the same meaning and they may be biased. Invariance checks include several elements [35,36]. First, each group (e.g., nulliparas vs. multiparas) should have the same pattern of indicators and factors (configural invariance). Second, factor loadings for the same indicators should be invariant across groups (metric invariance, also known as weak factorial invariance). Third, intercepts of the same items should be invariant across groups (scalar invariance, also known as strong factorial invariance). Fourth, residuals (errors) of the same items should be invariant across groups (residual invariance, also known as strict factorial invariance). Fifth, variances of the same factors should be invariant across groups (factor variance invariance). Sixth, means of factors should be invariant across groups (factor mean invariance). Elements 2–4 are called measurement invariance. Elements 5 and 6 are called structural invariance. Hypothesis testing should be conducted in the above sequence [36]. If one step is rejected, the next steps should not be performed.

The validity of a scale should be examined in terms of its construct validity. This means the degree to which the scale measures what it claims to measure. For example, the SDS can be considered valid if scores are significantly correlated with the severity of the mental and physical conditions tightly connected with social disability such as emesis, depression, and insomnia.

### Research Question

This study aims to examine measurement and structural invariance of the SDS among pregnant women and evaluate the correlation of SDS scores with scores for emesis and nausea, depression, and insomnia as external variables of construct validity.

## 2. Methods

### 2.1. Study Procedures and Participants

Approximately 1500 pregnant women at gestational weeks 10–13 were invited to participate in this study at the antenatal clinic of one general hospital and five private clinics located in Tokyo, Chiba, Ibaraki, and Kagoshima Prefectures in Japan. A set of questionnaires was distributed on two occasions, 1 week apart. We thought that a 1 week interval was appropriate because the participants’ mental state and nausea and vomiting would substantially change in reality rather than reflecting the (un)reliability of measurement if the interval was set at longer than this (such as a 2 week interval). The total sample consisted of 382 pregnant women, corresponding to a participation rate of approximately 25%. Of these, 129 women responded to the retest after 1 week. Test and retest responses were matched by a predetermined number on the set of questionnaires to assure anonymity. Pregnant women were excluded if they (a) were not fluent in Japanese, (b) were aged under 20 years, (c) had an eating disorder, (d) had vaginal bleeding or abdominal pain, (e) had a subchorionic hematoma, or (f) had experienced recurrent miscarriages. Although this was a convenience sample, it consisted of women receiving different types of obstetrical services in Japan. The study period was from January 2017 to May 2019.

### 2.2. Measurements

We used the Japanese version of the SDS [37]. This is a three-item self-report scale that measures disabilities in domains of (a) work and schoolwork, (b) social and leisure activities, and (c) family life and home responsibilities. Each item is rated from 0 to 10. Arbuckle et al. reported that Cronbach’s alpha coefficient was 0.89 and that the SDS total and item scores are significantly correlated with other measures, including the Global Assessment of Functioning [33].

The Japanese version [38] of the 24 h Pregnancy–Unique Quantification of Emesis and Nausea (PUQE-24) [39] was used simultaneously as a measure of nausea and vomiting during pregnancy. The PUQE-24 is a scoring system based on self-reporting of (a) nausea (duration of nausea in hours in the last 24 h), (b) vomiting (number of vomiting episodes in the last 24 h), and (c) retching (number of retching episodes in the last 24 h), each with a 5 point scale. Measurement invariance and validity were confirmed by Hada et al. [38].

The Japanese version [40,41] of the Patient Health Questionnaire-9 (PHQ-9) [42] was used simultaneously as a measure of depression. Each item checks for frequency of depressive symptoms over the previous 2 weeks with a four-point Likert scale from 0 to 3. The Japanese version of the PHQ-9 has a two-factor structure. The first factor consists of sleep change, fatigue, and appetite change items. The second factor consists of loss of interest, depressed mood, self-blame, concentration difficulty, psychomotor symptom, and suicidality items. Measurement invariance has been confirmed Wakamatsu et al. [43]. 

The Japanese version [44] of the Insomnia Severity Index (ISI) [45] was used simultaneously as a measure of insomnia. The ISI is a self-rating measure consisting of seven items with a five-point Likert type scale ranging from no problem (0) to very severe problem (4). The total score can range from 0 to 28. The Japanese version of the ISI has a two-factor structure. The first factor represents early, middle, and later insomnia and sleep dissatisfaction. The second factor represents interference of insomnia-induced difficulties with daytime functioning, noticeable sleep problems, and worry about sleep problems. Measurement invariance has been confirmed [46]. 

### 2.3. Data Analysis

First, we confirmed measurement invariance of the Japanese version of the SDS. As the SDS consists of only three items, it has a single-factor structure and we did not perform a series of exploratory factor analyses (EFAs). After calculating mean, SD, skewness, and kurtosis of each SDS item, we evaluated the factorability of the SDS using the Keiser–Meyer–Olkin (KMO) index and Bartlett’s sphericity test [47]. A single-factor EFA was sought for identification of factor loading for the SDS items. Next, confirmatory factor analysis (CFA) of the single-factor analysis was performed to evaluate goodness-of-fit. Model fit was examined in terms of chi-squared, comparative fit index (CFI), standardized root mean square residual (SRMR), and root mean square error of approximation (RMSEA). A good fit was defined as *χ*^2^/*df* < 2, CFI > 0.97, SRMR < 0.05, and RMSEA < 0.05. An acceptable fit was defined as *χ*^2^/*df* < 3, CFI > 0.95, SRMR < 0.10, and RMSEA < 0.08 [48,49]. 

The configural, measurement, and structural invariances of each model were examined across parity and observation time. Measurement invariance is to evaluate the hypothesis of equal loadings and, additionally, equal thresholds [35,36]. Multi-group confirmatory factor analysis (MGCFA) is one of the techniques to evaluate it. Starting with configural invariance, we assessed metric, scalar, residual, and factor variance invariances before assessing factor mean invariance [36,50,51]. Progress from one step to the next was judged as acceptable if (a) the *χ*^2^ decrease was not significant for the *df* difference, (b) the decrease in CFI was less than 0.01, or (c) the increase in RMSEA was less than 0.01 [52,53] We applied this procedure because a *χ*^2^ decrease is strongly sensitive to the sample size (*N*). Particularly with large samples, it can result in an unreasonable rejection of the invariance test.

To assess the construct validity of the SDS, we calculated the correlation between SDS scores and PUQE-24, PHQ-9, and ISI scores. We hypothesized that PUQE-24, PHQ-9, and ISI subscales could impact SDS scores. Thus, we performed a series of hierarchical multiple regression analyses. We entered demographic variables (age, parity, and gestational age) in the first step. Next, we entered the PUQE-24 total score, PHQ-9 subscale scores, and ISI subscale scores in the second step.

### 2.4. Ethical Considerations

This study was approved by the ethics committees of the Kitamura Institute of Mental Health Tokyo (No. 2015052301) and Kagoshima University (No. 170247).

## 3. Results

The mean (SD) age of the participants was 31.9 (4.9) years. The mean (SD) age of their partners was 33.4 (5.4) years. Unmarried women were rare (6%). Of the participants, 43.9% were nulliparas and 54.8% were multiparas. All participants reported neither current depression episode nor general anxiety disorder. Women who reported a current manic episode, and insomnia were one and one, respectively. Those cases were excluded from analyses. A total of 377 of those had no pharmacotherapy (e.g., antidepressant, antianxiety, or sleep medications), 3 were missing. Mean, SD, skewness, and kurtosis of each SDS item are shown in Table 1. The three items are mildly positively skewed (0.50 to 0.98) and kurtosis is low (−0.62 to 0.25). KMO is 0.735 and the chi-square statistic (*df*) is 618.901 (3), *p* < 0.001 from Bartlett’s test of sphericity. Therefore, the data appear factorable. Factor loadings of the items in a single-factor model are high, ranging from 0.79 to 0.90. CFA of this single-factor model shows good fit with the data: *χ*^2^/*df* = 0.000, CFI = 1.000, SRMR = 0.000, and RMSEA = 0.520.

Configural and measurement invariances between the nulliparas and multiparas and between test and retest occasions are acceptable (Table 2). Factor means are also similar between the primiparas and multiparas, as well as between test and retest occasions (Table 3).

SDS, PUQE-24, PHQ-9, and ISI scores are moderately correlated with each other (Table 4). After controlling for demographic variables in the first step, the SDS total score is significantly predicted by the PUQE-24, PHQ-9, and ISI scores (*F* [8] = 39.830, *p* < 0.001). *R*^2^ changes at step 2 is 0.476 (Table 5). All of these variables are significantly correlated with SDS scores. This model explains 47% of the variance in SDS scores (adjusted *R*^2^ = 0.47).

## 4. Discussion

This study shows that the single-factor structure of the SDS is robust in pregnant Japanese women. Its structure is invariant regardless of parity or observation time. Taking into account the SDS’s brevity and simplicity, we think that the use of the SDS in clinical and research settings for perinatal mental health care is promising. This is particularly the case when clinicians and researchers observe pregnant women’s disabilities across mental states such as depression, anxiety, fear, and insomnia, as well as physical complaints such as low body weight, pain, and skin conditions. The SDS may be used as an outcome measure for interventions by midwives and other perinatal health care professionals.

Disability in early pregnancy can be substantially explained by NVP, depression, and insomnia. Nearly half of the variance in QOL of pregnant women is explained by these symptoms. NVP impairs women’s QOL and, therefore, their ability to maintain day-to-day activities as well as work capacity in the domains of work/schoolwork, social/leisure, and family life/home responsibility [14]. According to a systematic review of factors influencing the quality of life [17], NVP, depression, and sleep disturbance are significantly associated with the poor quality of life. In this line, our results of hierarchical multiple regression analyses explain the poor quality of life. It is a characteristic condition in pregnant women. However, it is likely that the burdens or disabilities of pregnant women will not be noticed.

Disability is an umbrella term for impairments, activity limitations, and participation restrictions. It denotes the negative aspects of the interaction between an individual (with a health condition) and that individual’s contextual factors (environmental and personal factors) [54]. Therefore, various health-related conditions that have consequences of an impact on well-being in women’s life, need to be regarded as the disability. These physical and psychological symptoms should be carefully treated and attention should be paid to disability among pregnant women.

The different links between each of the three items with other correlates might be of interest to researchers. For example, Gutiérrez-Rojas et al. studied the predictors of the three SDS items in patients with bipolar disorder [55]. They found that work disability is predicted by the number of previous manic episodes and hospitalizations, as well as lower educational level; social life disability is predicted by the number of previous depressive episodes; and family life disability is predicted by higher age and drinking problems. The three SDS items during pregnancy may be predicted by different correlates and may predict different outcomes, including the course of pregnancy, early birth, and problems in neonates.

## 5. Limitations

This study has some limitations. The study was based on a convenient medium-sized sample. Generalizations should be made with caution. The inclusion criteria included mothers at gestational weeks 10–13. Although this was intended to include a homogeneous population of pregnant women, we may have had different results if different gestational weeks had been studied. The findings were based on self-report. Concordance of self-reports with data from clinical observers or family members should be examined.

## 6. Conclusions

Taking into consideration the drawbacks mentioned above, the SDS might be promising as an easy and robust measure of disability among pregnant women.

## Figures and Tables

**Table 1 healthcare-10-02514-t001:** Mean, SD, skewness, and kurtosis of SDS items (*N* = 377).

	Content	N	Mean	SD	Skewness	Kurtosis	Factor Loading Single-Factor Model
1	Work/school	377	3.1	2.6	0.50	−0.62	0.85
2	Social life	376	2.7	2.7	0.80	−0.15	0.90
3	Family life/home responsibilities	376	2.4	2.6	0.98	0.25	0.79

**Table 2 healthcare-10-02514-t002:** Measurement and structural invariance of the SDS.

	*χ^2^*	*df*	*χ*^2^/*df*	Δ*χ*^2^ (*df*)	CFI	ΔCFI	RMSEA	ΔRMSEA	AIC	Judgement
Nulliparas (*n* = 168) vs. multiparas (*n* = 208)										
Configural	0.000	0	0.000	Ref	1.000	Ref	0.042	Ref	36.000	ACCEPT
Metric	3.304	2	1.652	3.304 (2) NS	0.998	0.002	0.098	0.057	35.304	ACCEPT
Scalar	23.129	5	4.626	19.825 (3) *	0.971	0.027	0.084	+0.016	49.129	ACCEPT
Residual	28.980	8	3.786	5.851 (3) NS	0.966	0.005	0.081	+0.003	48.980	ACCEPT
Factor variance	31.255	9	3.601	2.276 (1) NS	0.964	0.002	0.373	0.289	49.255	ACCEPT
Time 1 (*n* = 380) vs. Time 2 (*n* = 128)										
Configural	0.000	0	0.000	Ref	1.000	Ref	0.000	Ref	36.000	ACCEPT
Metric	0.532	2	0.266	0.532 (2) NS	1.000	0.000	0.000	0.000	32.532	ACCEPT
Scalar	5.833	5	1.167	5.301(3) NS	0.998	0.002	0.024	0.024	31.833	ACCEPT
Residual	12.261	8	1.533	6.428 (3) NS	0.990	0.008	0.043	0.019	32.261	ACCEPT
Factor variance	12.405	9	1.378	0.144 (1) NS	0.992	+0.002	0.036	+0.007	30.405	ACCEPT

NS, not significant; * *p* < 0.001.

**Table 3 healthcare-10-02514-t003:** Factor mean invariance of the SDS.

	Differences in the Factor Mean (SE)
Multiparas as compared with nulliparas	0.070 (0.223) NS
Time 2 as compared with Time 1	0.056 (0.215) NS

NS, not significant; SE, standard error.

**Table 4 healthcare-10-02514-t004:** Correlation of the SDS, PUQE-24, PHQ-9 subscales, and ISI subscales.

	1	2	3	4	5	6
1: SDS total	−					
2: PUQE-24 total	0.39 *	−				
3: PHQ-9 1st factor	0.58 *	0.36 *	−			
4: PHQ-9 2nd factor	0.63 *	0.39 *	0.61 *	−		
5: ISI 1st factor	0.39 *	0.29 *	0.58 *	0.44 *	−	
6: ISI 2nd factor	0.41 *	0.29 *	0.51 *	0.37 *	0.62 *	−

* *p* < 0.001. *Note.* SDS, Sheehan Disability Scale; PUQE-24, Pregnancy–Unique Quantification of Emesis and Nausea; PHQ-9, Patient Health Questionnaire-9; ISI, Insomnia Severity Index.

**Table 5 healthcare-10-02514-t005:** Hierarchical multiple regression analyses of SDS with PUQE-24 and PHQ-9 subscales and ISI subscales in pregnancy.

	*R* ^2^	*R*^2^ *Change*	*F*	*F Change*	*df*	*Β*
Step 1. Demographic	0.003	Ref	0.355	Ref	3	
Age						0.049
Parity						0.014
Gestational age						−0.013
Step 2. NVP, depression, insomnia in pregnancy	0.479	0.476	39.830 ***	63.326	8	
PUQE-24						0.131 **
PHQ-9 1st factor						0.254 ***
PHQ-9 2nd factor						0.388 ***
ISI 1st factor						−0.023
ISI 2nd factor						0.022 *
Adjusted R^2^	0.47					

* *p* < 0.05; ** *p* < 0.01; *** *p* < 0.001. *Note.* SDS, Sheehan Disability Scale; PUQE-24, Pregnancy–Unique Quantification of Emesis and Nausea; PHQ-9, Patient Health Questionnaure-9; ISI, Insomnia Severity Index.

## Data Availability

The datasets used and analysed in the present study are available from the corresponding author upon reasonable request.

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
