# Peer review of "Disability during Early Pregnancy: Using the Sheehan Disability Scale during the First Trimester in Japan"

_healthcare, 2022, doi:10.3390/healthcare10122514_

Round 1

Reviewer 1 Report (New Reviewer)

Pregnancy is also characterized by difficulties related to physical problems such as nausea and vomiting, also known as hyperemesis gravidarum.  The sentence should be improved since pregnant women present nausea as a physiological mechanism and it does not necessarily cause the pathology called hyperemesis gravidarum.  

Whatever the cause, poor functioning should be seen as a disability.  Such disabilities are common  

I prefer to use the term difficulty, I don't think the pregnant woman has disabilities since they are normal women.  

Single women were rare  

This strange term should better describe it, causing confusion in the reader.

Author Response

Pregnancy is also characterized by difficulties related to physical problems such as nausea and vomiting, also known as hyperemesis gravidarum. The sentence should be improved since pregnant women present nausea as a physiological mechanism and it does not necessarily cause the pathology called hyperemesis gravidarum.

We deleted the sentence ‘also known as hyperemesis gravidarum.’ (p. 2, line 7)

Pregnancy is also characterized by difficulties related to physical problems such as nausea and vomiting, also known as hyperemesis gravidarum.

Whatever the cause, poor functioning should be seen as a disability. Such disabilities are common. I prefer to use the term difficulty, I don't think the pregnant woman has disabilities since they are normal women. Single women were rare 

This strange term should better describe it, causing confusion in the reader.

We added those sentences. (p.7, line 1~)

Disability is an umbrella term for impairments, activity limitations and participation restrictions. It denotes the negative aspects of the interaction between an individual (with a health condition) and that individual's contextual factors (environmental and personal factors) (WHO, 2001). 

Reviewer 2 Report (New Reviewer)

Congratulations! In the list of references I see only a few references after 2020. These are mainly linked to your own research. Could you provide more - non personal - references? 

Author Response

Congratulations! In the list of references I see only a few references after 2020. These are mainly linked to your own research. Could you provide more - non personal - references?

Following the reviewer’s comments, we added other references and discussed the term of ‘disability’ (p. 7, line 1~).

Reviewer 3 Report (New Reviewer)

The authors focused on disability during early pregnancy in Japan, and made validation of SDS as an assessment tool,Disability among pregnant women is one of the largest clinical problem and sometimes difficult to assess properly. However, the submitted manuscript does not deserve publication in the journal healthcare in its present form, mainly because of inappropriate writing. Please consider my comments below.

[Introduction]

L1. “little or no…” What era back are you referring to? Show reference to avoid an overstatement.

L3. The sentence “Regarding…” seems against the first two sentences; consider an appropriate conjunction to be comprehensive.

L8. “spurious” requires references.

L11. “similar” to what? Referring to antenatal and postnatal? Consider rewriting.

L16. As a perinatal healthcare worker, I feel artificiality to see eating disorders here. A prevalence of eating disorders would be smaller than those of depression, obsessive-convulsive disorder and sleep disorder. Additionally, eating disorder is preexisting (reference 4 deals with preexisting eating disorder among pregnant women) while others are induced by pregnancy. Consider rewriting, at least consider placing eating disorders after sleep disorders.

L36. A reference might be required to regard antenatal depression as common.

L37. “should cast” is an opinion of authors? If so, very inappropriate, and discuss later with reasoning. If referring to current situation of perinatal care worldwide, please show reference. Depression and NVP, and dysfunction due to them are of major problems perinatal healthcare workers are struggling with every day, as you should know.

L40. “regrettable” seems just an authors’ impression, and it is inappropriate to be here because introduction section would be a neutral basis of discussion. Keep neutral, although I understand what authors would like to say.

[Methods]

L5. I am afraid the interval of one week might be enough to deal with two occasions of testing as different. Please show some reason to set the interval to one week, with published reports if any.

L27. The reference number of 37 seems an error; should it be 39?

L47. The reference 47 has not been published; this is an inappropriate self-citation.

[Results]

L3. Considering that the prevalence rates of depression and general anxiety disorder are higher than that of manic psychosis, the study population including one mania without any depression or general anxiety disorder seems rather unnatural. In addition, the test result by a manic woman could have large effect on interpretation of the study results (although I think it is not so likely considering the sample size). Please show why the mania was not excluded.

[Discussion]

L4. “extremely” seems an overstatement. Keep objective.

L4. The authors conclude that SDS is “extremely” promising to observe pregnant women’s depression and anxiety, while the study population includes none of them. How do you guarantee SDS is valid for a population including them?

[Conclusion]

L2. The authors argue that perinatal care providers should pay more attention to the difficulties and disabilities of pregnant women, but there is no result to justify it. If the argument is true, the evidence that attention perinatal care providers are paying to pregnant women is insufficient should be presented. This study is not on that, so the authors can never make such a conclusion. A scientific validity is lacking.

[Overall]

-       The title is misleading because it does not show the content. The authors conducted a study just on validation of SDS among pregnant women, but did not study their disability.

-       Discuss the association of SDS and other scales.

-       I am afraid PUQE-24 is not yet in widespread use in Japan, so that it would be dangerous to use for validation.

-       Please include the Japanese version of SDS for possible extended studies, with reasoning of validity of the Japanese version if possible.

-       Is there any discussion on SDS as a screening tool of disability among first-trimester pregnant women?

-       Rererences 50 and 51 are not referred anywhere in the manuscript.

-       Some grammatical errors are found. Please correct them.

Author Response

L1. “little or no…” What era back are you referring to? Show reference to avoid an overstatement.

Following the succinct comment of the reviewer, we changed the L1 sentence as followed;

Pregnancy used to be considered to involve little or no psychological and physical difficulties.

L3. The sentence “Regarding…” seems against the first two sentences; consider the appropriate conjunction to be comprehensive.

We think that the reviewer correctly pointed out inappropriateness of the phraseology. We deleted “Regarding ….”.

L8. “spurious” requires references.

Yes. Again, we added a part to the following manner;

The argument that pregnancy is a psychologically stable period based only on the admission rate may be spurious and paradoxical because psychiatric admission is only an indicator of severe mental illness.

L11. “similar” to what? Referring to antenatal and postnatal? Consider rewriting.

Again, we changed the sentence as follows;

A recent epidemiological study showed that women are, during the ante-natal versus post-natal periods, at similar risk of developing mental illness, mainly mood and anxiety disorders.

L16. As a perinatal healthcare worker, I feel artificiality to see eating disorders here. A prevalence of eating disorders would be smaller than those of depression, obsessive-convulsive disorder and sleep disorder. Additionally, eating disorder is preexisting (reference 4 deals with preexisting eating disorder among pregnant women) while others are induced by pregnancy. Consider rewriting, at least consider placing eating disorders after sleep disorders.

Following the reviewer’s comment, we placed eating disorders after sleep disorders.

L36. A reference might be required to regard antenatal depression as common.

We added a reference.

L37. “should cast” is an opinion of authors? If so, very inappropriate, and discuss later with reasoning. If referring to current situation of perinatal care worldwide, please show reference. Depression and NVP, and dysfunction due to them are of major problems perinatal healthcare workers are struggling with every day, as you should know.

We added other references and discussed the term ‘disability’ (p. 7, line 1~).

L40. “regrettable” seems just an authors’ impression, and it is inappropriate to be here because introduction section would be a neutral basis of discussion. Keep neutral, although I understand what authors would like to say.

We revised the sentence as following;

Nevertheless, it is regrettable that less attention has been paid to this aspect of disability in pregnant women. Those impaired functions or functional difficulties were dealt with as burdens [14], also recognized as correlates of adverse pregnancy outcomes [19].

[Methods]

L5. I am afraid the interval of one week might be enough to deal with two occasions of testing as different. Please show some reason to set the interval to one week, with published reports if any.

We added our argument as follows;

We thought that a 1-week interval was appropriate because the participants’ mental state and nausea and vomiting would substantially change in reality rather than reflecting (un)reliability of measurement if the interval would be set at longer than this (such as a 2-week interval).

L27. The reference number of 37 seems an error; should it be 39?

We corrected numbering.

L47. The reference 47 has not been published; this is an inappropriate self-citation.

The 7th edition of APA Publication Manual (at page 336) notes that a manuscript in preparation (not yet published) should be cited with the year the author read it. Hence, we believe that this citation is appropriate.

[Results]

L3. Considering that the prevalence rates of depression and general anxiety disorder are higher than that of manic psychosis, the study population including one mania without any depression or general anxiety disorder seems rather unnatural. In addition, the test result by a manic woman could have large effect on interpretation of the study results (although I think it is not so likely considering the sample size). Please show why the mania was not excluded.

Indeed we wondered whether we should exclude a case of mania (which was self-reported, not interview based) but we thought such deletion would be arbitrary and thus this case remained.

[Discussion]

L4. “extremely” seems an overstatement. Keep objective.

We deleted ‘extremely’.

L4. The authors conclude that SDS is “extremely” promising to observe pregnant women’s depression and anxiety, while the study population includes none of them. How do you guarantee SDS is valid for a population including them?

We treated PHQ-9 subscales (i.e., sums of variables for Somatic symptoms and Non-somatic symptoms) in this study. Many studies showed that psychological symptoms are not categorical but dimensional in taxometric analyses (Haslam, 2003; Haslam et. al., 2020). Therefore, we did not treat depression as dichotomous variables (i.e., Yes or No) but as continuous variables. Of course, cases showing high scores of the PHQ-9, which indicated MDE was included in our study sample. We recognize that disabilities due to those symptoms in pregnancy are important to evaluate their individual quality of life. We need to further examine SDS or other useful measurement tools for disabilities during pregnancy.

References:

Haslam, N. (2003). Categorical versus dimensional models of mental disorder: the taxometric evidence. The Australian and New Zealand Journal of Psychiatry, 37(6), 696-704. https://doi.org/10.1080/j.1440-1614.2003.01258.x

Haslam, N., McGrath, M. J., Viechtbauer, W., & Kuppens, P. (2020). Dimensions over categories: A meta-analysis of taxometric research. Psychological Medicine, 50(9), 1418-1432.

[Conclusion]

L2. The authors argue that perinatal care providers should pay more attention to the difficulties and disabilities of pregnant women, but there is no result to justify it. If the argument is true, the evidence that attention perinatal care providers are paying to pregnant women is insufficient should be presented. This study is not on that, so the authors can never make such a conclusion. A scientific validity is lacking.

We deleted the part the reviewer pointed out.

[Overall]

The title is misleading because it does not show the content. The authors conducted a study just on validation of SDS among pregnant women, but did not study their disability.

We are unable to comprehend this comment. As noted a reply to the Reviewer 1, we measured disability experienced by pregnant women.

Discuss the association of SDS and other scales.

Yes, we did it.

I am afraid PUQE-24 is not yet in widespread use in Japan, so that it would be dangerous to use for validation.

Unfortunately, there is no other questionnaire to measure the severity of emesis or NVP which validated to use in Japan. Therefore, we used the PUQE-24 in this study.

Please include the Japanese version of SDS for possible extended studies, with reasoning of validity of the Japanese version if possible.

The content of the SDS is protected by copyright.

Is there any discussion on SDS as a screening tool of disability among first-trimester pregnant women?

As seen, our study is the first to use SDS among a Japanese expectant woman population.

Rererences 50 and 51 are not referred anywhere in the manuscript.

We corrected it.

Some grammatical errors are found. Please correct them.

We corrected it.

Round 2

Reviewer 1 Report (New Reviewer)

No

Author Response

L1. “little or no…” What era back are you referring to? Show reference to avoid an overstatement.

Following the succinct comment of the reviewer, we changed the L1 sentence as followed;

Pregnancy used to be considered to involve little or no psychological and physical difficulties.

This does not answer my comment. The authors seem unwilling to amend this sentence, since “was previously considered” and “used to be considered” have almost the same meaning. I understand the authors’ insistence, but it must be followed by an appropriate reference, especially at the beginning of the manuscript. Show reference(s) which refers to the fact that pregnancy used to be considered to involve little or no psychological or physical difficulties. Actually, I doubt it.

[Answer] Many studies treated physical disabilities of pregnant women, such as cerebral palsy and spinal cord injuries, are those associated with limits to mobility, flexibility, and dexterity; sensory disabilities, intellectual and developmental disabilities, such as Down syndrome, autism spectrum disorder, and foetal alcohol spectrum disorder (Horner-Johnson et. al., 2016; Horner-Johnson et. al., 2020; Tarasoff et. al., 2020). However, those disabilities differ from our intentions. Whereas NVP, insomnia, and various psychosocial problems were treated in many studies, pregnancy has been regarded as a natural condition. To our best knowledge, no study treated impairments, difficulties, and dysfunctions with (due to) pregnancy as ‘disabilities’. How on earth shall we present evidence that virtually nobody was interested in what did not attract reserchers at all?

References

Horner-Johnson, W., Darney, B. G., Kulkarni-Rajasekhara, S., Quigley, B., & Caughey, A. B. (2016). Pregnancy among US women: differences by presence, type, and complexity of disability. American Journal of Obstetrics and Gynecology, 214(4), 529-e1.

Horner‐Johnson, W., Dissanayake, M., Wu, J. P., Caughey, A. B., & Darney, B. G. (2020). Pregnancy intendedness by maternal disability status and type in the United States. Perspectives on Sexual and Reproductive Health, 52(1), 31-38.

Tarasoff, L. A., Ravindran, S., Malik, H., Salaeva, D., & Brown, H. K. (2020). Maternal disability and risk for pregnancy, delivery, and postpartum complications: A systematic review and meta-analysis. American Journal of Obstetrics and Gynecology, 222(1), 27-e1.

L8. “spurious” requires references.

Yes. Again, we added a part to the following manner;

The argument that pregnancy is a psychologically stable period based only on the admission rate may be spurious and paradoxical because psychiatric admission is only an indicator of severe mental illness.

This does not answer my comment. “Spurious” is a strong word, so that when used it should be followed by sufficient reasoning. The authors just deny the study by Paffenbarger et al., only based on their own ideas. This would not be justified. Please revise the sentences appropriately.

[Answer] We are not with the reviewer that authors should not make comments on their OWN ideas.

L37. “should cast” is an opinion of authors? If so, very inappropriate, and discuss later with reasoning. If referring to current situation of perinatal care worldwide, please show reference. Depression and NVP, and dysfunction due to them are of major problems perinatal healthcare workers are struggling with every day, as you should know.

We added other references and discussed the term ‘disability’ (p. 7, line 1~).

This does not answer my comment. Keep neutral and avoid the authors’ dogmatic ideas in the introduction section. You can insist something in the discussion section with relevant results of your study.

[Answer] We deleted the sentence “Perinatal health care professionals should cast light on these issues.”

[Methods]

L5. I am afraid the interval of one week might be enough to deal with two occasions of testing as different. Please show some reason to set the interval to one week, with published reports if any.

We added our argument as follows;

We thought that a 1-week interval was appropriate because the participants’ mental state and nausea and vomiting would substantially change in reality rather than reflecting (un)reliability of measurement if the interval would be set at longer than this (such as a 2-week interval).

Comment: As an obstetrician I think one week is not sufficient for impaired mental state and nausea and vomiting of pregnant women to resolve.

[Answer] Devellis (2017) described four factors that are confounded when the examines two sets of scores on the same measure, separated in time. These are (a) real change in the construct of interest (e.g., a net increase in the average level of anxiety among a sample of individuals) (b) systematic oscillations in the phenomenon (e.g., variations in anxiety around some construct mean as a function of time of a day) (c) changes attributable to differences in subjects or measurement methods rather than the phenomenon of interest (e.g., fatigue effects that cause items to be misread), and (d) temporal stability due to the inherent unreliability of the measurement procedure. Only the fourth factor is unreliability. Physical and psychological conditions in early pregnancy are changeable with the progress of gestation. We thought that a 1-week interval was appropriate to avoid strictly those confounds.

Reference

Devellis, F. R. (2017). Scale development: Theory and applications. 4th edition. Sage publishing.

[Results]                                                                                                    

L3. Considering that the prevalence rates of depression and general anxiety disorder are higher than that of manic psychosis, the study population including one mania without any depression or general anxiety disorder seems rather unnatural. In addition, the test result by a manic woman could have large effect on interpretation of the study results (although I think it is not so likely considering the sample size). Please show why the mania was not excluded.

Indeed we wondered whether we should exclude a case of mania (which was self-reported, not interview based) but we thought such deletion would be arbitrary and thus this case remained.

Comment: I am afraid the inclusion/exclusion criteria was not well considered.

[Answer] We reconsidered the sample for this study. The criteria you pointed out are right. Therefore, we re-analysed all statistics, using the sample from which was excluded a case with current manic episode and a case with current insomnia. The results were not changed. Please review on results tables.

[Overall]

The title is misleading because it does not show the content. The authors conducted a study just on validation of SDS among pregnant women, but did not study their disability.

We are unable to comprehend this comment. As noted a reply to the Reviewer 1, we measured disability experienced by pregnant women.

As the authors themselves mention in the conclusion section, this study is on the validity and usability of SDS as a measurement of disability of pregnant women, with comparison to other indicators (questionnaires). The tentative title would lead readers to think this study would answer a question of how disabled Japanese pregnant women are in the first trimester, but neither is the aim of the study to answer it nor does the conclusion section include the answer. Please reconsider the title.

[Answer] As noted previous reply, we measured disabilities experienced by pregnant women. We indicated that the latent construct of disability measured by the SDS was explained by nausea and vomiting, depression, and insomnia among women in early pregnancy, in this study. That’s not by frequencies but by rates of explanation of the psychological phenomenon. We believe that resolving what explains a psychological phenomenon is important. Therefore, the title of this article is entitled ‘Disability during early pregnancy: using the Sheehan Disability Scale during the first trimester in Japan’.

Reviewer 3 Report (New Reviewer)

I am glad to review a revised version.

L1. “little or no…” What era back are you referring to? Show reference to avoid an overstatement.

Following the succinct comment of the reviewer, we changed the L1 sentence as followed;

Pregnancy used to be considered to involve little or no psychological and physical difficulties.

This does not answer my comment. The authors seem unwilling to amend this sentence, since “was previously considered” and “used to be considered” have almost the same meaning. I understand the authors’ insistence, but it must be followed by an appropriate reference, especially at the beginning of the manuscript. Show reference(s) which refers to the fact that pregnancy used to be considered to involve little or no psychological or physical difficulties. Actually, I doubt it.

L8. “spurious” requires references.

Yes. Again, we added a part to the following manner;

The argument that pregnancy is a psychologically stable period based only on the admission rate may be spurious and paradoxical because psychiatric admission is only an indicator of severe mental illness.

This does not answer my comment. “Spurious” is a strong word, so that when used it should be followed by sufficient reasoning. The authors just deny the study by Paffenbarger et al., only based on their own ideas. This would not be justified. Please revise the sentences appropriately.

L37. “should cast” is an opinion of authors? If so, very inappropriate, and discuss later with reasoning. If referring to current situation of perinatal care worldwide, please show reference. Depression and NVP, and dysfunction due to them are of major problems perinatal healthcare workers are struggling with every day, as you should know.

We added other references and discussed the term ‘disability’ (p. 7, line 1).

This does not answer my comment. Keep neutral and avoid the authors’ dogmatic ideas in the introduction section. You can insist something in the discussion section with relevant results of your study.

[Methods]

L5. I am afraid the interval of one week might be enough to deal with two occasions of testing as different. Please show some reason to set the interval to one week, with published reports if any.

We added our argument as follows;

We thought that a 1-week interval was appropriate because the participants’ mental state and nausea and vomiting would substantially change in reality rather than reflecting (un)reliability of measurement if the interval would be set at longer than this (such as a 2-week interval).

Comment: As an obstetrician I think one week is not sufficient for impaired mental state and nausea and vomiting of pregnant women to resolve.

[Results]

L3. Considering that the prevalence rates of depression and general anxiety disorder are higher than that of manic psychosis, the study population including one mania without any depression or general anxiety disorder seems rather unnatural. In addition, the test result by a manic woman could have large effect on interpretation of the study results (although I think it is not so likely considering the sample size). Please show why the mania was not excluded.

Indeed we wondered whether we should exclude a case of mania (which was self-reported, not interview based) but we thought such deletion would be arbitrary and thus this case remained.

Comment: I am afraid the inclusion/exclusion criteria was not well considered.

[Overall]

The title is misleading because it does not show the content. The authors conducted a study just on validation of SDS among pregnant women, but did not study their disability.

We are unable to comprehend this comment. As noted a reply to the Reviewer 1, we measured disability experienced by pregnant women.

As the authors themselves mention in the conclusion section, this study is on the validity and usability of SDS as a measurement of disability of pregnant women, with comparison to other indicators (questionnaires). The tentative title would lead readers to think this study would answer a question of how disabled Japanese pregnant women are in the first trimester, but neither is the aim of the study to answer it nor does the conclusion section include the answer. Please reconsider the title.

Author Response

L1. “little or no…” What era back are you referring to? Show reference to avoid an overstatement.

Following the succinct comment of the reviewer, we changed the L1 sentence as followed;

Pregnancy used to be considered to involve little or no psychological and physical difficulties.

This does not answer my comment. The authors seem unwilling to amend this sentence, since “was previously considered” and “used to be considered” have almost the same meaning. I understand the authors’ insistence, but it must be followed by an appropriate reference, especially at the beginning of the manuscript. Show reference(s) which refers to the fact that pregnancy used to be considered to involve little or no psychological or physical difficulties. Actually, I doubt it.

[Answer] Many studies treated physical disabilities of pregnant women, such as cerebral palsy and spinal cord injuries, are those associated with limits to mobility, flexibility, and dexterity; sensory disabilities, intellectual and developmental disabilities, such as Down syndrome, autism spectrum disorder, and foetal alcohol spectrum disorder (Horner-Johnson et. al., 2016; Horner-Johnson et. al., 2020; Tarasoff et. al., 2020). However, those disabilities differ from our intentions. Whereas NVP, insomnia, and various psychosocial problems were treated in many studies, pregnancy has been regarded as a natural condition. To our best knowledge, no study treated impairments, difficulties, and dysfunctions with (due to) pregnancy as ‘disabilities’. How on earth shall we present evidence that virtually nobody was interested in what did not attract reserchers at all?

References

Horner-Johnson, W., Darney, B. G., Kulkarni-Rajasekhara, S., Quigley, B., & Caughey, A. B. (2016). Pregnancy among US women: differences by presence, type, and complexity of disability. American Journal of Obstetrics and Gynecology, 214(4), 529-e1.

Horner‐Johnson, W., Dissanayake, M., Wu, J. P., Caughey, A. B., & Darney, B. G. (2020). Pregnancy intendedness by maternal disability status and type in the United States. Perspectives on Sexual and Reproductive Health, 52(1), 31-38.

Tarasoff, L. A., Ravindran, S., Malik, H., Salaeva, D., & Brown, H. K. (2020). Maternal disability and risk for pregnancy, delivery, and postpartum complications: A systematic review and meta-analysis. American Journal of Obstetrics and Gynecology, 222(1), 27-e1.

L8. “spurious” requires references.

Yes. Again, we added a part to the following manner;

The argument that pregnancy is a psychologically stable period based only on the admission rate may be spurious and paradoxical because psychiatric admission is only an indicator of severe mental illness.

This does not answer my comment. “Spurious” is a strong word, so that when used it should be followed by sufficient reasoning. The authors just deny the study by Paffenbarger et al., only based on their own ideas. This would not be justified. Please revise the sentences appropriately.

[Answer] We are not with the reviewer that authors should not make comments on their OWN ideas.

L37. “should cast” is an opinion of authors? If so, very inappropriate, and discuss later with reasoning. If referring to current situation of perinatal care worldwide, please show reference. Depression and NVP, and dysfunction due to them are of major problems perinatal healthcare workers are struggling with every day, as you should know.

We added other references and discussed the term ‘disability’ (p. 7, line 1~).

This does not answer my comment. Keep neutral and avoid the authors’ dogmatic ideas in the introduction section. You can insist something in the discussion section with relevant results of your study.

[Answer] We deleted the sentence “Perinatal health care professionals should cast light on these issues.”

[Methods]

L5. I am afraid the interval of one week might be enough to deal with two occasions of testing as different. Please show some reason to set the interval to one week, with published reports if any.

We added our argument as follows;

We thought that a 1-week interval was appropriate because the participants’ mental state and nausea and vomiting would substantially change in reality rather than reflecting (un)reliability of measurement if the interval would be set at longer than this (such as a 2-week interval).

Comment: As an obstetrician I think one week is not sufficient for impaired mental state and nausea and vomiting of pregnant women to resolve.

[Answer] Devellis (2017) described four factors that are confounded when the examines two sets of scores on the same measure, separated in time. These are (a) real change in the construct of interest (e.g., a net increase in the average level of anxiety among a sample of individuals) (b) systematic oscillations in the phenomenon (e.g., variations in anxiety around some construct mean as a function of time of a day) (c) changes attributable to differences in subjects or measurement methods rather than the phenomenon of interest (e.g., fatigue effects that cause items to be misread), and (d) temporal stability due to the inherent unreliability of the measurement procedure. Only the fourth factor is unreliability. Physical and psychological conditions in early pregnancy are changeable with the progress of gestation. We thought that a 1-week interval was appropriate to avoid strictly those confounds.

Reference

Devellis, F. R. (2017). Scale development: Theory and applications. 4th edition. Sage publishing.

[Results]                                                                                                    

L3. Considering that the prevalence rates of depression and general anxiety disorder are higher than that of manic psychosis, the study population including one mania without any depression or general anxiety disorder seems rather unnatural. In addition, the test result by a manic woman could have large effect on interpretation of the study results (although I think it is not so likely considering the sample size). Please show why the mania was not excluded.

Indeed we wondered whether we should exclude a case of mania (which was self-reported, not interview based) but we thought such deletion would be arbitrary and thus this case remained.

Comment: I am afraid the inclusion/exclusion criteria was not well considered.

[Answer] We reconsidered the sample for this study. The criteria you pointed out are right. Therefore, we re-analysed all statistics, using the sample from which was excluded a case with current manic episode and a case with current insomnia. The results were not changed. Please review on results tables.

[Overall]

The title is misleading because it does not show the content. The authors conducted a study just on validation of SDS among pregnant women, but did not study their disability.

We are unable to comprehend this comment. As noted a reply to the Reviewer 1, we measured disability experienced by pregnant women.

As the authors themselves mention in the conclusion section, this study is on the validity and usability of SDS as a measurement of disability of pregnant women, with comparison to other indicators (questionnaires). The tentative title would lead readers to think this study would answer a question of how disabled Japanese pregnant women are in the first trimester, but neither is the aim of the study to answer it nor does the conclusion section include the answer. Please reconsider the title.

[Answer] As noted previous reply, we measured disabilities experienced by pregnant women. We indicated that the latent construct of disability measured by the SDS was explained by nausea and vomiting, depression, and insomnia among women in early pregnancy, in this study. That’s not by frequencies but by rates of explanation of the psychological phenomenon. We believe that resolving what explains a psychological phenomenon is important. Therefore, the title of this article is entitled ‘Disability during early pregnancy: using the Sheehan Disability Scale during the first trimester in Japan’.

This manuscript is a resubmission of an earlier submission. The following is a list of the peer review reports and author responses from that submission.

Round 1

Reviewer 1 Report

The current study represents using the Sheehan Disability Scale during the first trimester in Japan.

Abstract should be rebuilt because of the shortness. Not enough information for the first opinion of readers.

Authors wrote about mental problems during the pregnancy, but have omitted the patients, who had it before. Maybe some data concerning women taking pharmacotherapy and having a pregnancy might be also important in the context of the study.

Please provide the excluded criteria for the current study too.

Methodology should be more detailed described.

Discussion is too short and do not reflect a deep analysis of received data.